# Mesenchymal Chondrosarcoma from Diagnosis to Clinical Trials

**DOI:** 10.3390/cancers15184581

**Published:** 2023-09-15

**Authors:** Monika Dudzisz-Śledź, Monika Kondracka, Monika Rudzińska, Agnieszka E. Zając, Wiktoria Firlej, Dorota Sulejczak, Aneta Borkowska, Bartłomiej Szostakowski, Anna Szumera-Ciećkiewicz, Jakub Piątkowski, Piotr Rutkowski, Anna M. Czarnecka

**Affiliations:** 1Department of Soft Tissue/Bone Sarcoma and Melanoma, Maria Sklodowska-Curie National Research Institute of Oncology, 02-781 Warsaw, Poland; monika.dudzisz-sledz@pib-nio.pl (M.D.-Ś.); monika7373@onet.pl (M.K.); mrudzinska19@gmail.com (M.R.); agnieszka.zajac@pib-nio.pl (A.E.Z.); bartlomiej.szostakowski@pib-nio.pl (W.F.); aneta.borkowska@pib-nio.pl (A.B.); bartek@szostakowski.pl (B.S.); piotr.rutkowski@pib-nio.pl (P.R.); 2Faculty of Medicine, Medical University of Warsaw, 02-091 Warsaw, Poland; 3Department of Experimental Pharmacology, Mossakowski Medical Research Centre Polish Academy of Sciences, 02-106 Warsaw, Poland; dots@op.pl; 4Department of Pathology, Maria Sklodowska-Curie National Research Institute of Oncology, 02-781 Warsaw, Poland; szumann@gmail.com; 5Department of Diagnostic Hematology, Institute of Hematology and Transfusion Medicine, 02-776 Warsaw, Poland; 6Institute of Genetics and Biotechnology, Faculty of Biology, University of Warsaw, 02-106 Warsaw, Poland; j.piatkowski@biol.uw.edu.pl

**Keywords:** mesenchymal chondrosarcoma, rare sarcomas, treatment, prognosis

## Abstract

**Simple Summary:**

Mesenchymal chondrosarcoma (MCS) is a subtype of chondrosarcoma with rare occurrence and poor survival rates. MCS stains positive for S-100 and SOX9 as well as CD99, ezrin, and NKX2.2. Recurring fusion of the *HEY1* and *NCOA2* genes—involved in epigenetic modifications—was reported in MSC. MCS may also be positive for *IRF2BP2-CDX1* fusion, loss of the cyclin-dependent kinase inhibitor 2A (*CDKN2A*)/*p16* or loss of *TP53*. Treatment of these tumors is difficult and there is lack of therapeutical options for patients with advanced and metastatic disease due to the unknown pathogenesis of MCS. Despite the limited efficacy of conventional chemotherapy in an advanced setting, young patients may be considered for chemotherapy combined with aggressive local treatment and/or RT. The data of other therapeutic options, including immunotherapy efficacy, are limited. If available, patients with MCS should be considered potential candidates for clinical trials.

**Abstract:**

Mesenchymal chondrosarcoma (MCS) is a rare subtype of chondrosarcoma with a poor prognosis. Although these tumors are sensitive to radiotherapy/chemotherapy, the standard treatment for localized MCS is only surgical resection, and there are no established treatment guidelines for patients with advanced and metastatic MCS. Due to the low incidence of MCS, the pathology of these tumors is still unknown, and other therapeutic options are lacking. Some studies show the potential role of the PDGF/PPI3K/AKT, PKC/RAF/MEK/ERK, and pRB pathways, and BCL2 overexpression in the pathogenesis of MCS. These findings provide an opportunity to use protein kinases and BCL2 inhibitors as potential therapy in MCS. In this review, we summarize the current knowledge about MCS diagnosis and treatment options. We show the immunological and molecular biomarkers used in the diagnosis of MCS. In addition, we discuss the known prognostic and predictive factors in MCS. Finally, we present the novel trends, including targeted therapies and ongoing clinical trials using protein kinase inhibitors and the death receptor 5 (DR5) agonist, which may be the focus of future MCS treatment studies.

## 1. Introduction

Mesenchymal chondrosarcoma (MCS) is a very rare type of sarcoma, accounting for approximately 1–10% of all chondrosarcomas (CS) [1] in children and adolescents [2]. MCS is a high-grade and malignant mesenchymal tumor with a biomorphic histological pattern of well-differentiated cartilage and small round cells [3,4]. MCS probably originates from immature chondroblasts, differentiating into well-differentiated cartilage, often with enchondral ossification [3]. MCS usually occurs in young adults in the second and third decades of life, while most patients with conventional chondrosarcomas (CCS) are over 50 years old [5]. The incidence rate of MCS in men and women is similar [4]. According to a November 2015 retrospective review of chondrosarcoma subtypes using the SEER database, the mesenchymal subtype effects the largest proportion of African Americans (16.4%) and Asian individuals (about 9.3%) [6,7,8].

MCS is considered a separate entity completely different from CCS. The difference is mainly in incidence rate, clinical course, and rarity. MCS tends to be more aggressive than CCS, with a tendency toward late recurrences. Its 5-year survival rate is around 55% [6], and almost 20% of the cases had metastatic disease at diagnosis [3]. MCSs are very rare tumors—only 226 cases of MCS were reported in the Surveillance, Epidemiology, and End Results (SEER) database between 1973 and 2013 [4,5]. The German Society for Pediatric Oncology and Hematology (GPOH) identified 15 patients with MCS who participated in the studies carried out by two study groups, cooperative soft tissue sarcoma (CWS) and osteosarcoma (COSS), the Study Groups of the German Society for Pediatric Oncology and Hematology (GPOH). The median age of these 15 patients was 16.6 years (range 1.4–25.2). There was less than 0.2% with MCS in the more than 7000 patients registered with sarcoma [5].

Little is known about its pathogenesis because of the rare occurrence of MCS. MCS treatment is limited due to its high malignancy and metastatic potential, so new potential therapeutic targets are needed [9]. In this paper, we summarize the current state of knowledge on the pathogenesis and treatments of MCS. We also present potential therapeutic targets with a summary of the currently ongoing clinical trials for MCS.

## 2. Chondrosarcoma Diagnostics

### 2.1. Clinical Presentation

MCS can be distributed in bone, soft tissue, or intracranial sites [3]. Among bones, the most frequent are craniofacial regions, ribs, ilium, spine, and femur [3]. Intrathecal MCS of the spinal cord most often affects the upper lumbar and lower thoracic spine. These tumors are usually solitary and usually located on the right side of the spine [1]. Approximately one-third of MCS is found in extraskeletal soft tissues, such as the central nervous system, and the meninges are one of the most common locations [1]. These cases are called extraskeletal MCSs (EMCs). The sites most frequently affected by EMC are the head and neck, followed by the lower extremities. However, there are rare reports of EMC involving various soft tissue and visceral locations [10], but the female genitalia, kidneys, pancreas, chest wall, and retroperitoneal space are rare sites of origin for EMC [11,12]. A study conducted in 2021 at the Department of Radiology of the Memorial Sloan Kettering Cancer Center in New York included 23 patients (average age 28 ± 6 years) presenting 13 skeletal and 10 EMCs. Regarding the location of the tumors, eleven cases affected the limbs (47.8%), four thoracic or chest wall (17.4%), four were in the head and neck (17.4%), three in the pelvis (13%), and one in the retroperitoneal space (4.4%). In terms of bone or soft cell origin, 13 cases (56.5%) were in the bones, while the observed cases (43.5%) were extraskeletal and localized in soft tissues. It should be noted that there were no age or gender differences in the distribution of skeletal and EMC [13,14].

There have also been discovered such cases as of dumbbell-shaped intramedullary MCS with isolated punctate calcification (only 7 cases of this type have been described since 1978) [15], secondary pancreatic MCS (a total of 13 cases have been reported) [4], primary renal EMC (only 16 patients with renal EMC have been described) [14], primary orbital intraocular MCS (40 cases reported in the literature) [16], and secondary thyroid MCS (only one case of this metastases has been reported in the literature) [17]. Nevertheless, MCS commonly present as large masses (>10 cm) and often present with chondroid calcifications (80%). In the study by Ghafoor et al. [13], the average size of the MCSs, measured in the longest dimension, was 10.2 ± 7.2 cm (0.6–21.3 cm). The mean tumor size in non-metastatic cases was 7.1 ± 7.3 cm (0.6–25 cm), while the average tumor size for metastases was 13.2 ± 5.9 cm (6.9 to 23 cm, *p* = 0.004) [13].

### 2.2. Symptoms and Signs 

Due to the different locations of MCSs, their symptoms may be various and non-specific. In general, patients with MCS seek medical attention due to pain and other symptoms caused by compression of the lesion. However, a diagnosis of MCS is often delayed due to nonspecific signs and symptoms and non-oncology unit referrals [14]. The first signs and symptoms are typically associated with tumor mass and are often described as painless masses in the specific localization. In patients with intracranial MCS, the most common symptoms are diplopia caused by compression of the intracranial nerve (III, IV, VI), visual field defects, and headaches [18]. In addition, vomiting and limb convulsions may occur [19]. A limited number of mediastinal MCS case reports describe nonspecific symptoms such as inspiratory pain, cough, and shortness of breath. Intraspinal tumors have been associated with back pain similar to this patient; however, these may have focal neurologic deficits dependent on the spinal level compressed or damaged [12]. In general, MCS tends to present with some type of pain.

### 2.3. Imagining

Imaging plays an important role in the accurate treatment and diagnosis of MCS. Based solely on imaging diagnostics, there is a high probability that MCS is recognized as a different tumor, as CT scans are non-specific [1]. The main imaging methods in the diagnosis of MCS include cases, X-ray, computed tomography (CT), and magnetic resonance imaging (MRI), as well as positron emission tomography (PET). Imaging should be followed by biopsy [14]. In CT, chondroid mineralization is usually seen, and the lesion may appear heavily calcified. However, ‘finely stippled’ calcification may be more common. On MRI, compared to other conventional chondrosarcomas, MCS has a different pattern of contrast enhancement. The diffuse and typical chondroid septal and peripheral enhancement is often absent. A feature not shown in other chondrosarcomas but presented in MCS, is low-signal, serpentine, and high-flow vessels in some areas. In conclusion, the picture suggestive of the diagnosis of MCS is an aggressive osseous lesion with subtle chondroid matrix mineralization and an intermediate signal on T2WI, which is lower than in conventional chondrosarcoma [20] (Figure 1). In magnetic resonance imaging, most MCSs are isointense at T1WI and T2WI and tend to be lobulated with well-defined boundaries. The tumor mass is hypointense in T1WI and hyperintense in T2WI, which differs significantly from typical isointense manifestations. On X-ray images, MCS usually appears as a soft tissue mass or osteolytic lesion with indistinct or well-defined boundaries, and induration is often present. In PET, MCS can be detected by technetium 99-methylene diphosphonate bone scan (99 m Tc-MDP), while a biopsy is performed in doubtful cases to facilitate diagnosis and subsequent treatment [14]. In general, the diagnosis of MCS is difficult, and the final diagnosis is mainly based on histopathology, with multiple immunohistochemistry staining as described below [14]. MCS tends to metastasize with a high rate of recurrence (RR), so at minimum chest and abdomen imaging is needed in these cases [15]. Staging is very important during the primary diagnosing of MCS because metastases and large tumor size contribute to reduced survival and need multidisciplinary treatment [4].

### 2.4. Histopathology

The characteristic histological features of MCSs were described by Lichtenstein et al. [14]. Histopathological MCSs are characterized by bidirectional differentiation of tumor cells consisting of alternating small round or spindle-shaped undifferentiated mesenchymal cells, which aggregate into dense groups, with some isolated cells and islands of hyaline cartilage (Figure 2). The small undifferentiated cells have a round or oval nucleus with a thick nuclear membrane, granular chromatin, modest nucleolus, and scanty cytoplasm. Mitosis in this area of the tumor is frequent and often atypical, and necrosis and inflammation may be present [21,22]. On the contrary, cells in the cartilage part of the tumor are polygonal, with vacuoles in the cytoplasm, a round nucleus with prominent granular chromatin, and a well-distinguished nucleolus. Solid areas of small round cells have been described as “Ewing-like,” and spindle-shaped cells arranged around blood vessels have been called “hemangiopericytoma” or “hemangiopericytoma-like” [23]. The cartilaginous and undifferentiated areas are well-defined or intertwined as they migrate, and hemangiopericytoma-like changes can be seen in the undifferentiated areas [23,24].

Occasionally, in the absence of the characteristic biphasic morphology (for example, due to sampling bias on biopsy), the differential diagnosis of other cartilaginous tumors or small round cell tumors may be challenging in pathology [13]. Therefore, in differential diagnosis, it is necessary to distinguish MCS particularly accurately from tumors such as meningioma, solitary fibrous tumor/hemangiopericytoma, oligodendroglioma, chordoma, poorly differentiated synovial sarcoma, lymphoma, small cell osteosarcoma, embryonal rhabdomyosarcoma, and other types of CS [25].

Immunohistochemical detection of the corresponding biomarkers can help distinguish MCS from other tumors of similar appearance. MCS cells are known to express mesenchymal markers. These include vimentin, CD99, B-cell lymphoma (BCL-2), and SRY-Box Transcription Factor 9 (SOX9) [23,26]. The expression of these markers and CD57 is detected in the small cells of many MCSs, while the S-100 protein is found in their cartilaginous regions [26].

The expression of particular markers can also help differentiate the mesenchymal and other subtypes of CS. MCS stains positive for S-100 and SOX9, like CCS [27]; however, it tends to be CD99, ezrin, and NKX2.2 positive, in contrast to CCS [28,29]. On the other hand, similarly to other subtypes of CS, MCS may demonstrate p53 overexpression [30]. In the study by Park et al. [31], nuclear positivity of p53 was observed within a mean of 37.3% in tumor cells. Additionally, positive staining of the mesenchymal component was observed. It was found in over 80% of analyzed tumor tissues [32].

Furthermore, MCSs small cell areas show higher expression of catenin beta-1 (CTNB1) compared to its cartilaginous components and low-grade CS, for example, clear cell CS [29]. A study by van Oosterwijk et al. [33] revealed that MCS cells could also stain positive for nuclear SMAD Family Member 1 (SMAD1), involved in transforming growth factor beta (TGFβ)/bone morphogenetic protein (BMP), in the small cell component of these tumors [31]. On the other hand, the cartilaginous component of MCS was positive for the enzyme that interacts with exostoses (EXT)1/2—N-deacetylase/N-sulfotransferase (NDST1) in half of the MCS analyzed. At the same time, it was highly expressed in the small cell component (in almost each MCS). In addition to highly expressed markers, there are also those in MCS that are not expressed in these tumors. One such marker is New York esophageal squamous cell carcinoma 1 (NY-ESO-1), expressed in CCS and dedifferentiated CS [33]. Other markers which have been reported, negative for MCS, are integration 1 of the Friend leukemia virus (FLI-1), smooth muscle actin (SMA), glial fibrillary acid protein (GFAP), and keratins [34,35].

However, testing for the presence of these markers in tumor cells alone may not be sufficient. It has been shown that the presence in MCS of undifferentiated small round cells, positive for CD99 immunohistochemistry in a membranous pattern, and also positive for NK2 homeobox 2 (NKX2.2) in a nuclear pattern, may be a pitfall as Ewing sarcoma (EWS), another relatively frequent primary bone tumor, is also positive for these markers. However, MCS differs from EWS as the only bone tumor with a component of developing cartilage, which stains positive for the S-100 protein [12,36]. Similarly, although SOX9 is considered a discriminatory marker that is positive in both cartilaginous and small cell areas and can distinguish MCS from other small cell tumors, it is not a specific marker. It can be detected in other cartilaginous tumors. In contrast, the lack of expression of MyoD1 and Myogenin can differentiate rhabdomyosarcoma from MCS, whereas INI1 (integrase interactor 1) expression separates MCS from atypical teratoid tumors [35]. Furthermore, STAT6 is considered a highly sensitive marker of solitary fibrous tumor (SFT), and the lack of (a signal transducer and activator of transcription-6 (STAT6) expression can be used to differentiate MCS from SFT [14].

Therefore, new specific markers are constantly being sought for the differential diagnosis of MCS. One candidate is oligodendrocyte lineage transcription factor 2 (OLIG2), which Yao et al. indicates as present in 12 of the 14 MCS cases analyzed [24]. OLIG2 is a transcription factor that plays an important role during the development of the central nervous system. It is detected in proper brain parenchyma and brain tumors [37]. These include medulloblastoma, gliomas, central neurocytomas, and glio-neuronal tumors [38,39,40,41]. Interestingly, co-expression of OLIG2 and SOX9 was observed in part of the MCS [42]. Yoshida and colleagues demonstrated the usefulness of NKX3.1 as an MCS marker [43]. However, other researchers have not confirmed this finding. Therefore, for a proper diagnosis, it is necessary to supplement tumor imagining, histological, and immunohistochemical studies with extensive molecular analysis [44,45].

## 3. Genetic Alterations and Potential Targets for Novel Therapies

The genetics of MCS are still poorly understood due to the low frequency of these subtypes among all CSs diagnosed. However, some genetic abnormalities have already been described, such as hairy/split-enhancer-related with a fusion of the YRPW motif 1—nuclear receptor coactivator 2 (*HEY1*-*NCOA2*) fusion [46]. The most common gene alterations in bone and soft tissue MCSs are different gene fusions for each of them, demonstrating the genetic heterogeneity of this CS subtype [47,48]. Selected molecular alterations have been known in MCS for a long time [46] and are used in diagnosis. Some of the studies presented below revealed the potential pathways activated in the pathogenesis of these tumors, which could become novel candidates for future therapies (Table 1).

### 3.1. HEY1-NCOA2 Fusion and Related Signaling Pathways

In 2012, Wang et al. showed their interest in the recurring fusion of the *HEY1* and *NCOA2* genes on chromosome 8(q13; q21), which appeared to be involved in MCS tumorigenesis [46]. The same fusion was also found in the study by Nyquist et al. in three MSC cases originating from bone tissue [47]. Qi et al. used iPSC-derived mesenchymal stem cells (iPSC-MSCs) to functionally characterize the fusion protein in MCS [57]. The researchers generated stably transduced iPSC-MSCs with inducible expression of the HEY1-NCOA2 fusion protein, wild-type HEY1, or wild-type NCOA2 and proved that the HEY1-NCOA2 fusion protein significantly improves cell proliferation. This occurs by preferential binding of the HEY1-NCOA2 fusion to promoter regions of the HEY1 protein targets and transcriptional activation by recruiting coactivators through its NCOA2 component [57].

HEY-NCOA2 may also be involved in epigenetic modifications, such as chromatin remodeling, through the recruitment of coactivator elements and histone methyltransferases by the AD1 and AD2 domains of NCOA2. Therefore, histone deacetylase inhibitors (HDACi) may be a promising candidate for future therapies in MCS [58]. In a study by Tanaka et al. [59], treatment with the HDACi—Panobinostat—caused suppression of MSC cells growth both in vitro and in vivo. Panobinostat downregulated the expression of genes downstream of HEY1-NCOA2 and Runx2 [59]. HDACi has been approved in treatment of many other tumors, including sarcomas, for example Romidepsin, Belinostat, and others [51]. Belinostat in combination with guadecitabine or cedazuridine has been tested in a phase II trial study (NCT04340843) on patients with unresectable and metastatic CS, but it was recently suspended.

Additionally, wild-type HEY1 is considered a downstream effector of the Notch signaling pathway. It reduces gene expression levels of the alpha 1 chain of type II collagen (*COL2A1*), which encodes an essential component of the cartilaginous extracellular matrix [58,60]. Furthermore, the HEY1-NCOA2 fusion gene directly targets and upregulates both the platelet-derived growth factor receptor Alpha (*PDGFRA*) and Beta (*PDGFB*), dramatically increasing the level of phospho-protein kinase B (AKT) (Ser473) [57]. These findings suggest exploring PDGF/phosphatidylinositol 3-kinase (PI3K)/AKT as a potential target for the treatment of MCS. Furthermore, the same study identified direct transactivation of *BCL2* by the HEY1–NCOA2 fusion protein and upregulation of hairy and enhancer of split-1 (*HES1*) and *SOX4*, which participate in the chondrocyte differentiation process [57].

Recently, TQB3525, an PI3K inhibitor, has been tested for advanced bone sarcomas with PI3KA mutations (NCT04690725). In several studies, the anti-apoptotic BCL2 protein was overexpressed in MCS; therefore, BCL2 inhibitors, e.g., Venetoclax used in hematological malignancies treatment [52], may be a promising target in MCS treatment [33,57,61]. This finding was confirmed in research by de Jong et al., who indicate that the use of inhibitors of the proteins of the BCL2 family restores the apoptotic machinery of the MCS cell line and sensitizes it to conventional chemotherapy [62]. Strong expression of PDGFRA, together with protein kinase C alpha (PKC-α), leading to BCL2 phosphorylation and antiapoptotic activity, was also observed in malignant mesenchymal chondroblasts [60]. The role of the PKC pathway in MCS may confirm the fact that CD99 (expressed in MCS as mentioned in Section 4) plays a role in the mediation of activation of the mitogen-activated protein kinase (MAPK)/extracellular signal-regulated kinase (ERK) and MAPK/Jun N-terminal kinase (JNK) pathways through the PKC pathway mentioned previously, which may also be a promising target for future therapies [61].

### 3.2. IRF2BP2-CDX Fusion and Other Cytogenetic Changes

Using FISH and whole transcriptome sequencing analysis, Nyquist et al. [47] found a new case of MCS that demonstrated a t(1; 5)(q42; q32) translocation as a unique karyotypic aberration. It results in the fusion of the interferon regulatory factor 2 (*IRF2BP2*)-caudal type homeobox 1 (*CDX1*) gene between exon 1 of the *IRF2BP2* gene on chromosome 1 and intron 1 of the *CDX1* gene on chromosome 5. *CDX1* belongs to the homeobox gene family and encodes a transcription factor protein that upstreams the expression of Hox genes, which are also known to be involved in different types of malignancies, such as leukemias [63]. This novel fusion was detected in one of the four MCSs included in the studies. MCS cases show that the *IRF2BP2-CDX1* fusion originated in soft tissue [47]. Furthermore, the MCS positive for *IRF2BP2-CDX1* did not show *HEY1-NCOA2* fusion, presented by cases originating from bone tissue, which can demonstrate the heterogeneity of MCS genetics [47]. According to the presence of *IRF2BP2-CDX1* fusion only in one case until now and the limited data, further studies are necessary on a larger population.

Furthermore, in one case of a young patient with MCS of bone, t(11; 22)(q24; q12) translocation was observed, the same as that observed in EWS and peripheral neuroectodermal tumor [64,65]. This translocation causes Ewing Sarcoma Breakpoint Region 1 (*EWSR1*) and Friend leukemia integration 1 transcription factor (*FLI1*) gene fusion, which is a diagnostic marker in EWS and can be a promising target for future therapies [66]. Presently, phase 1 and 2 clinical trials using Lurbinectedin in FET (FUS, EWRS1, TATA-Box-Binding Protein Associated Factor 15 (TAF15))-Fusion Tumors are ongoing (NCT05918640). Moreover, targeting facilitates the chromatin transcription (FACT) complex, directly induced by EWSR1-FLI1, which may be a promising direction in the treatment of tumors with this genetic aberration, e.g., by FACT inhibitor—CBL0137 [53]. Others, less common, are translocations involving chromosomes 13 and 21 (der(13; 21)(q10; q10)), reported in two cases of MCS originating from both bone and soft tissue [67], or aberrations in chromosome 7 (add (p13)) and 22 (add (q13)), identified in young patients [68].

### 3.3. Other Reported Genetic Changes

Not much is known about other genetic changes beyond the fusions mentioned above. Another mutation, reported in MCS by Meijer et al., is the loss of the cyclin-dependent kinase inhibitor 2A (*CDKN2A*)/*p16*, which is involved in the retinoblastoma protein (pRB) pathway, observed in three of nine analyzed cases [32]. Inactivation of *CDKN2A* leads to activation of cyclin-dependent kinase (CDK); therefore, CDK inhibitors may be a promising strategy for the treatment of p16-defective tumors [54]. CDK inhibitors—Palbociclib 1, Ribociclib 2, and Abemaciclib 3 have been already approved for ER+/HER2- breast cancer treatment [69]. Currently, phase 2 clinical trial using Abemaciclib in CS is ongoing (NCT04040205). The researchers also indicated a gain in the entire 12 chromosomes (in six out of nine cases) and a hemizygous loss of *TP53* (in two of nine cases) [32]. Another study indicated a possible deletion of *TP53* in 6 of 33 MCS samples, suggesting that genetic alterations of this gene can occur, although they are not common events [31]. Currently, many potential drugs against p53 mutant are ongoing, e.g., Zoledronic acid in triple negative breast cancer (NCT03358017), Lamivudine in colorectal cancer (NCT03144804), and ATO (arsenic trioxide/Trisenox) in acute myeloid leukemia (NCT03855371), etc. [70].

A study by Zehir et al. [71] involved sequencing in 10,000 patients with different advanced tumors, including three cases of MCS. The researchers indicated four genes with point mutations, each detected in one out of three cases [71]. These genes encoded the androgen receptor (AR), insulin receptor (INSR), alpha-thalassemia/mental retardation, X-linked (ATRX) chromatin remodeler, and the patched protein homolog (PTCH1) involved in the Hedgehog pathway (Hh) for bone development and chondrocyte differentiation [72]. The Hh inhibitor, Vismodegib, has already been approved in basal cell carcinoma treatment [73]; however, clinical trial on advanced CS with vismodegib did not meet the primary end point of this trial [55]. INSR, as a tyrosine kinase receptor, is involved in the Ras/mitogen-activated protein kinase (MAPK) and PI3K/AKT pathways, which may be a promising area for future research [74]. Nowadays, FDA-approved mitogen-activated protein kinase kinase (MEK) inhibitors, e.g., in melanoma or non-small-cell lung cancer treatment, are Trametinib, Cobimetinib, Selumetinib, or Binimetinib [56]. However, several clinical trials using drugs, e.g., D3S-002 or QLH11906, targeting advanced solid tumors with MAPK pathway mutations are presently ongoing (NCT05886920, NCT05488821).

According to available data, the presented genetic alterations and involved signaling pathways, especially protein kinases and antiapoptotic proteins, may be promising targets for future therapies. However, there is still insufficient data, and further research in this area is highly recommended.

### 3.4. Diagnostic Genetic Markers

Over the years, a fusion of the HEY1-NCOA2 gene has been successfully used as a molecular diagnostic marker for MCS, helping to diagnose cases lacking typical histological features [4,25,26,75,76]. Many studies suggest that tumors with nested round cell morphology and staghorn vasculature lacking a distinctive cartilaginous component should undergo FISH testing for HEY1-NCOA2 fusion [75,77]. Another well-known diagnostic marker in MCS is the lack of isocitrate dehydrogenase (*IDH*)*1/2* mutations, common in central, periosteal, and dedifferentiated CS [7,32]. Until now, no other markers commonly used in diagnostics are known, and further research is strongly encouraged.

## 4. Radical Surgical Treatment

Surgical resection remains the mainstay of MCS treatment for individuals with localized disease. Compared with other subtypes, MCS represents more aggressive behavior and worse survival outcomes [6,7,8]. Surgery should always aim at resection with disease-free margins. Unfortunately, due to anatomic constraints, obtaining wide margins is not always possible, and, in some cases, amputation is needed (Figure 1). Therefore, sometimes neoadjuvant or adjuvant therapy has to be considered; non-radical resection results in a high risk of local recurrence [7,78].

Despite MCSs being slow-growing tumors, the local recurrence risk is high [79] and may occur many years after primary treatment [80]. According to Strach et al. [7], the time interval between initial treatment and local RR may extend to 20 years, indicating the need for long-term surveillance in the management [7,9]. Many studies underline the importance of negative surgical margins in reducing the local recurrence rate [6,7,9]. However, Xu et al. [6] are the first to demonstrate a significant association between surgical margins and the final survival rate, including relapse-free survival (RFS) and event-free survival (EFS). In their study, negative margins were significantly associated with improved local RFS (*p* < 0.001) and EFS (*p* = 0.050) [6]. According to the literature, postoperative radiation can remarkably improve local-recurrence-free survival for patients with positive margins [2,6,81]. However, it does not benefit the OS and EFS rates. Nevertheless, radiation therapy (RT) is recommended as the salvage therapy to achieve better local control without a clear surgical margin [6]. The literature analysis reveals that perioperative chemotherapy may also decrease the local recurrence rate in the localized setting [6,7,9]. However, the data concerning the possible survival benefit of chemotherapy remain inconsistent. Some have reported a reduction in the risk of death [9], while others demonstrate a lack of association with OS [6,81]. Moreover, studies evaluating perioperative chemotherapy in similar disease groups, such as EWS, have demonstrated survival advantages [7]. According to the study by Xu et al., the gender, origin, or site of the tumor is not significantly associated with OS or EFS [6].

In MCS, patients are usually younger than those with classic types, with a significant proportion of tumors being in axial or extraosseous locations. As postulated by Mody et al. [82], surgical management of tumors in spinal locations represents a significant challenge (Figure 3). Although en bloc-wide resection delivers the best solution for local tumor control, this technique has several limitations, including neurovascular anatomy and patient selection. Intralesional resection may help to paliate neurological symptoms but provides poor local control and is not recommended. According to Mendenhall et al., patients with R1 resected MCS or those not amenable to complete resection should be considered for adjuvant or definitive RT [78]. Nevertheless, radical R0 resection of mesenchymal chondrosarcomas is the gold standard for treatment, especially in spinal location [83]. Microsurgical technique should cover the initial debulking followed by the removal of residual tissue up to normal margins free from ChSa [84].

## 5. Radiation Therapy

RT should be considered in selected cases of MCS, mainly as adjuvant treatment when margins are positive, in nonresectable tumors, or in palliative patients [81,85,86]. Several studies suggest that MCS should be considered for RT, as this subtype tends to be radiosensitive [81,85,86]. However, a relatively high dose of RT is needed for the assumption of radical treatment. In conventional fractionations of 1.8–2 Gy, the total dose of RT should be greater than 60 Gy (60–72 Gy). However, in the tissues that are difficult for radical wide excision, RT also has dose-limits organs. Because of that, in areas that are difficult to target with a sufficiently high dose of RT due to limitations related to organs at risk, treatment with particles such as protons or heavy ion therapy can be considered. As delivering a dose higher than 70 Gy is mandatory to obtain local control of ChSa, modern techniques such as intensive modulated radiotherapy (IMRT), stereotactic radiosurgery, or finally hadron therapy should be considered [87,88]. The cyberknife system with hypofractionated doses was described in single reports an effective treatment in these tumors, alongside Gamma knife radiosurgery which has also been reported as a useful djuvant strategy after surgery with local control up to 80% after 5 years [89,90]. For the spine, skull base, and sinonasal mesenchymal ChSa, proton beam therapy may also be administered for gross tumor mass as well as microscopic residual disease after surgery [91,92].

Due to the aggressiveness and high rate of local recurrence of MCS, adjuvant RT may be considered an effective therapy option. However, there have been no clear findings in the literature, with some papers indicating a significant increase in survival and others not. In cases with inadequate margins and contaminated local spaces, most authors recommend adjuvant RT as a salvage therapy to achieve better local control [81]. Adjuvant RT at a dose of 60 Gy provides a high local control rate. However, there is no benefit in OS [8]. RT should be applied in unresectable and palliative cases to alleviate local symptoms, such as pain or bleeding. In such cases, hypofractionated radiotherapy is preferred, providing the patient with a shorter treatment time and more convenience [8].

The study by De Amorim Bernstein et al. evaluated the results of surgery and RT (median dose 60 Gy) in a group of 23 patients with local MCS [86]. The median OS for the total cohort of patients was 21.65 years, and the disease-free survival (DFS) rates at 3 and 5 years were 70.7% and 57.8%, respectively, without an obvious correlation between radiation dose and local tumor control. However, this study is relatively small and lacks a control group, making it difficult to draw statistical conclusions from its results [86]. On the other hand, Kawaguchi et al. [81] performed an analysis of 28 patients who had non-metastatic disease, 10 of whom received RT. The local recurrence-free survival was estimated using Kaplan–Meier plots. By the logarithmic rank test (*p* = 0.037), there was a statistically significant difference between the groups that received and did not receive RT; RT was associated with improved local-recurrence-free survival [81]. At the same time, Wand et al. evaluated 3759 patients with all types of CSs, divided into four groups: surgery (76.8%), surgery with adjuvant RT (10.7%), RT alone (3.0%), and without treatment (9.5%). After balancing the covariates to control for biased impact effects, there was no significant difference in OS and cancer-specific survival between the surgery group and the surgery and the RT group. However, the median OS was longer in the surgery group. The shortest OS was presented in the RT group. The findings generally indicate that patients with MCS who underwent surgical resection have a good prognosis and that RT does not prolong survival [85]. Finally, in the Australian study of twenty-two MCS patients, eleven received RT. Half of the patients with localized disease received RT, mostly (7 patients) in an adjuvant regimen. RT was one of the factors that showed a tendency to prolong OS but without statistical significance. All patients with primary metastatic disease received RT in combination with other treatments [7].

## 6. Treatment of Locally Advanced Disease

The standard treatment for localized MCS is surgical resection, and surgical margins are important in managing MCS [93,94]. The most common methods currently used in MSC, sparing surgery, include, in some tumor locations only, radical bone resections (shoulder, pelvis), but in reference sarcoma centers modular oncology endoprostheses (megaprostheses), non invasive growing prostheses used in children, bone auto and allografts, rotationplasties, patient specific surgical implants, and arthrodesis of large joints are also used [95]. At the same time, MCSs are poorly differentiated tumors sensitive to RT and chemotherapy [96]. Given the relatively small number of patients with CSs, it is recommended to refer MSC patients to reference sarcoma centers [93,94,97]. In our study by Rutkowski et al., 10-year OS decreased to 17% in patients who could not undergo radical excision [96].

MSC may be treated multidisciplinary with neoadjuvant chemotherapy, followed by radiotherapy and surgery as per current guidelines from both the National Comprehensive Cancer Network (NCCN), the European Society of Medical Oncology (ESMO), and also Polish guidelines relevant for our country [98,99,100,101,102]. In general, chemotherapy is not widely used in CSs, except in MCS, where the same regimens as in EWS are usually used, and dedifferentiated CS, where treatment can be combined with chemotherapy, including doxorubicin. For MCS, typically, the majority of multidisciplinary teams recommend surgery, chemotherapy, and RT, with an initial chemotherapy strategy similar to EWS and other soft tissue sarcomas. It was reported that adjuvant anthracycline-based chemotherapy for MSC is associated with a significant reduction in recurrence and death. The consensus of sarcoma experts is to use Ewing sarcoma–like treatment regimens in MSC, although no randomized trial has been performed to confirm this approach [103]. MSC systemic treatment therefore generally includes alternating cycles of etoposide plus ifosfamide (EI) and adriamycin with vincristine and cyclophosphamide (CAV) [93].

According to Mendenhall et al. [78], adjuvant RT is employed for patients with close (<5 mm) or positive margins and those with incompletely resectable tumors. However, it should be emphasized that data about RT for MCS are limited, and it is necessary to extrapolate data from soft tissue sarcomas and other bone sarcomas [78]. A similar study by De Amorim Bernstein et al. evaluated the results of surgery and adjuvant RT (median dose 60 Gy) as described above [86].

To analyze multidisciplinary treatment, Strach et al. [7] published the treatment results of 22 patients with MCS between 2001 and 2022 at Australian sites. They found that 19 of the 22 patients (86%) had localized disease at diagnosis, and 16 were treated with surgery (84%, surgery alone in 8 patients); 11 patients received RT (58%, 1 neoadjuvant and 7 adjuvant), and 10 received systemic chemotherapy (53%, 3 neoadjuvant, 2 adjuvant and 1 definitive). Recurrence of the disease and/or distant metastasis developed in 10 (52%) patients after primary treatment, and among them 9 occurred after 2 years of follow-up [7]. The median overall survival (OS) was 104.1 months (95% CI 25.8–182.3). There was improved OS in patients with localized MCS who underwent surgical resection of the primary tumor (*p* = 0.003) and patients in a better general performance status (ECOG 0–1 vs. 2–3, *p* = 0.023) [7]. On the other hand, Cesari et al. conducted a study on 21 patients with MCS who achieved complete surgical remission and they reported a DFS rate of 76% in those who received chemotherapy (9 patients) and 17% in those who did not receive chemotherapy (12 patients) [104]. The study published by Frezza et al., based on data from 113 patients with MCS, confirmed the reduced risk of recurrence and death in patients treated with chemotherapy [9]. This study’s median progression-free survival (PFS) and OS were 7 and 20 years, respectively. Systemic chemotherapy in localized disease was associated with a reduced risk of recurrence (*p* = 0.046; HR = 0.482 95% CI: 0.213–0.996) and death (*p* = 0.004; HR = 0.445 95% CI: 0.256–0.774). The risk of local recurrence was also lower in the cases of clear resection margins (2% vs. 27%; *p* = 0.002) and lack of distant metastases at diagnosis (*p* < 0.0001). Survival was not influenced by the primary site and origin of the tumor [9]. The data presented above confirm that the addition of chemotherapy improves DFS in resectable MCS. On the other hand, in the study by Xu et al., the treatment of chemotherapy improved EFS (*p* = 0.046) but did not benefit OS (*p* = 0.139) [6]. The differences in outcomes result from the disease’s rarity and the understandable lack of prospective trials. Although the potential benefit of adjuvant chemotherapy remains unclear, medically fit patients should be considered for anthracycline-based adjuvant chemotherapy [78].

### 6.1. Systemic Therapy in an Unresectable and Metastatic Setting

Patients with unresectable or metastatic disease will mainly undergo palliative treatment with no intention of cure. For patients with MSC metastatic disease, the prognosis is poor. The National Comprehensive Cancer Network (NCCN) guidelines and the European Society for Medical Oncology (ESMO) recommend following the Ewing Sarcoma Guidelines for MCS treatment. Due to a lack of prospective trials evaluating chemotherapy in CS patients, most results are derived from retrospective studies [105,106]. One of the studies evaluating the role of chemotherapy in advanced MCS was performed by Italiano et al. The research was based on the medical records of 180 patients with unresectable and/or metastatic CS who had histologically proven conventional, dedifferentiated, mesenchymal, or clear-cell CS and received first-line chemotherapy. The study revealed that 54.5% of patients received combination chemotherapy, and 45.5% were treated with single-agent chemotherapy. The majority of patients (73%) received an anthracycline-containing regimen. The results of the retrospective analysis showed that the objective response rate (ORR) based on RECIST was significantly different according to histological subtype, with the best response being observed for MCS (31%) and dedifferentiated CS (20.5%). ORR was also higher for patients who received combination therapy than those treated with single agents, although this difference did not reach significance. In multivariate analysis, single agent and combination chemotherapy did not correlate with improved OS. The results indicate the generally limited efficacy of conventional chemotherapy in patients with advanced CSs, with the best response in MCS and dedifferentiated CS [55]. Similar conclusions come from the study by Xu et al. [6]. Their systematic review of 107 patients with MCSs revealed that chemotherapy based on anthracyclines improves PFS but does not benefit OS [6,78,104].

Currently, no established treatment guidelines exist for patients with advanced and metastatic CS, resulting in very diverse treatment regimens used in practice, especially in second and further lines of treatment [106]. A retrospective comparison of different treatment regimens for 25 patients with unresectable MCS in four major sarcoma centers showed that multidrug chemotherapy based on doxorubicin is in favor and results in longer PFS than other regimens based on different chemotherapeutics [107,108]. Italiano et al. published the results of patients with unresectable and/or metastatic CS, including MCS (15 pts, 9.5%), who were treated with first-line systemic treatment with a cytotoxic agent, starting between 1988 and 2011 (*n* = 163), in 15 European and American institutions. Among the patients with MCS, ORR was 31% [55]. MCS generally responds to conventional chemotherapy more often than other CS subtypes, but the response may differ from case to case. This difference may be related to histology. The data for patients treated with chemotherapy alone are limited due to the multimodality treatment of this disease.

Many case studies and case reports confirm positive outcomes in patients with relapsed or metastatic MCS. One of those studies reported a good response to adjuvant chemotherapy in a patient with MCS of the buttocks and multiple pancreas, bone, and lung metastases [109]. After undergoing tumor resection and pancreatectomy, the patient received doxorubicin-based chemotherapy with a good response from metastatic tumors. At the end of therapy, most metastases were no longer detectable on positron emission tomography. However, there was a relapse in metastatic tumors 1 year and 3 months after surgery, and additional chemotherapy did not further suppress the tumor growth [109]. Another study by Araki et al. showed a long-lasting response of 12-month PFS in patients with relapsed MCS in the pelvis in a patient that received trabectedin monotherapy as a fourth-line chemotherapy [110]. Trabectedin should be considered, particularly as a second-line therapy, for treating translocation-related soft tissue sarcomas [111,112]. The clinical benefits of trabectedin were shown in a sub-analysis of a phase 2 study [55], which compared the outcomes of trabectedin and the best supportive care (BSC) treatments for MCS; they failed or were intolerable to standard chemotherapy [111,112]. The study was carried out in a small number of patients: two patients with extraskeletal myxoid chondrosarcoma (EMCS), three patients with MCS in the trabectedin group, and three patients with MCS in the BSC group. However, the results would justify this choice of treatment for patients with advanced EMCS or MCS who did not respond or were intolerable to standard chemotherapy [112]. The median PFS was 12.5 months (95% CI 7.5 to not reached) in the trabectedin group, while it was 1.0 months (95% CI 0.3 to 1.0 months) in subjects with MCS in the control group [112].

In 2008 Dantonello et al. [5] published the results of the treatment of 15 patients with mesenchymal sarcoma, 4 osseous and 11 extraosseous, with a median age of 16.6 (1–25) and a median follow-up of 9.6 years (1–22). All patients underwent resection, but complete resection was possible in only 8 of them. In the study, 13 patients received cytotoxic chemotherapy and 6 received RT. The 10-year event-free and OS rates were 53% and 67%, respectively [5]. The cytotoxic drugs used for the treatment of these patients were dactinomycin, carboplatin, cisplatin, cyclophosphamide, doxorubicin, etoposide, ifosfamide, high-dose methotrexate, and vincristine in various combinations [5]. In the study by Strach et al., recurrence treatment included surgical management with lung metastasectomy, RT in seven patients, and chemotherapy in five [7]. Multimodal treatment was applied in four patients. In sum, thirteen patients were treated with different chemotherapy regimens, mainly based on anthracycline (92%), but generally the same as used for EWS (doxorubicin, vincristine, cyclophosphamide, ifosfamide, etoposide and irinotecan with temozolomide) and soft tissue sarcomas (doxorubicin ± ifosfamide ± dactinomycin/dacarbazine/vincristine, gemcitabine with docetaxel). The patients also received immunotherapy and tyrosine kinase inhibitors (TKI) (pazopanib) and also participated in clinical trials [7].

We are waiting for the results of a prospective investigator-initiated, phase II clinical study MSC (NCT04305548) by Italian Sarcoma Group which explores the activity of trabectedin in a population of patients aged ≥16 years with progressive advanced (locally advanced or metastatic). Publication of the results from a phase 2, single arm, multi center trial evaluating the efficacy of the combination of sirolimus and cyclophosphamide in metastatic or unresectable myxoid liposarcoma and chondrosarcoma (NCT02821507) sponsored by Leiden University Medical Center is also anticipated.

### 6.2. Drugs in Development

The use of TKI in patients with advanced and metastatic sarcoma who do not respond to conventional therapy is currently being investigated [113]. At this point in time, we are waiting for the results from a pazopanib neoadjuvant trial in non-rhabdomyosarcoma soft tissue sarcomas (PAZNTIS): a phase ii/iii randomized trial of preoperative chemoradiation or preoperative radiation plus or minus pazopanib (NSC# 737754) which also recruited MCS patients. There is currently a study exploring pazopanib in patients with fibrosarcoma and EMCS (NCT02066285). A phase 1/2 study evaluated the combination of nivolumab with the mTOR inhibitor (rapamycin mammalian target) ABI-009 in patients with advanced malignancies, including CSs (NCT03190174). The data from the CS patient group are not available [113]. The expression of programmed cell death/programmed death ligand 1 (PD/L1) is found in approximately 50% of dedifferentiated CSs [113]. Evidence of the efficacy of immune checkpoint inhibitors in CSs, including the mesenchymal subtype, is still limited. The data on immunotherapy efficacy in specific MCS are very limited and the reported results are not very satisfactory [114]. Paoluzzi et al. retrospectively analyzed the efficacy of nivolumab in patients with metastatic sarcoma, including two with CS [115]. In one patient with MCS, the disease was stable after four cycles of nivolumab. More often, data are based on the response assessment in general in sarcomas or CSs. In the multicenter phase II SARC028 trial in patients with advanced soft tissue and bone sarcomas treated with pembrolizumab in one of five patients, the objective response to immunotherapy was reported [116].

Due to the rarity of MCSs, the literature available on immunotherapy among patients with advanced MCS is limited. Moreover, one of the preclinical studies revealed the absence of PD-L1 expression in six MCSs, suggesting that MCS is less suited for PD-1/PD-L1 blockade. Nevertheless, this cannot be generalized to all CSs [7]. Another study evaluating the effects of immunotherapy on patients with metastatic sarcoma was performed by Paoluzzi et al. [115]. The retrospective research was based on the clinical data of twenty-eight patients with metastatic or locally advanced soft tissue or bone sarcoma treated with nivolumab with or without pazopanib. Clinical benefit, including a partial response or a stabilization of disease, was observed in 50% of patients after at least four cycles of nivolumab. The observations provide a rationale for further exploring the efficacy of nivolumab and other checkpoint inhibitors [115].

For targeted therapies, case-based data for MSC is available. Vismodegib was studied in a phase 2 trial for Treating Patients with Advanced Chondrosarcomas (NCT01267955). Furthermore, regorafenib (NCT02048371) was studied in a phase II SARC024: trial. A study with INBRX, which is a tetravalent death receptor 5 (DR5) agonist antibody (NCT04950075), in unresectable or metastatic CS is currently underway [116]. Another ongoing phase 2 trial involves the study NCT02982486, which is dedicated to assessing the efficacy of nivolumab with ipilimumab in unresectable sarcoma and endometrial cancer. Lenvatinib in combination with pembrolizumab is currently being evaluated in a phase 2 non-randomized trial in patients with different advanced sarcomas (NCT04784247), including osteosarcomas and CSs [116]. The response of MSC to cabozantinib was also reported [117].

## 7. Conclusions

MCS is a rare and malignant mesenchymal tumor. MCSs can be located in bone, soft tissue, or intracranial sites, but during diagnosis, the high malignancy and metastatic potential of MCSs should be considered. The best option for confirming the diagnosis of MCS is imaging diagnostics, including MRI, CT, PET, and, in doubtful cases, X-ray and biopsy. However, imaging findings are not specific to MCS, so it is crucial to rely on histopathology, with immunohistochemistry as an additional tool.

The molecular pathology of MCS is still poorly understood, and only some genetic aberrations are known. The most common, used in routine diagnosis, is the presence of a *HEY1-NCOA1* fusion. Other genetic alterations include *IRF2BP2-CDX* and deletions of *CDKN2A* and *TP53*. Recent studies have indicated that MCS pathogenesis may involve the PDGF/PPI3K/AKT, PKC/RAF/MEK/ERK, and pRB pathways, as well as BCL2 overexpression, which may be a promising target for future therapies.

Due to a strong tendency of MCS towards local recurrence and distant metastases many years after primary treatment, the necessity of long-term surveillance should be emphasized. Negative surgical margins prove to be significantly associated with improved survival rates, including RFS and EFS. Without clear surgical margins, RT should be implemented as a salvage therapy. According to several studies, perioperative chemotherapy may also reduce local recurrence, but its possible survival benefit remain unclear. In the case of distant metastases, treatment should be determined individually based on the patients’ age and general state of health. Despite the limited efficacy of conventional chemotherapy in an advanced setting, young patients may be considered for chemotherapy combined with aggressive local treatment with surgery and/or RT. In contrast, elderly patients should be treated with palliative intent.

In summary, there are no established guidelines for the treatment of patients with advanced and metastatic MCS. The data of other therapeutic options, including immunotherapy efficacy, are limited and not successful enough. The rarity of these tumors makes the search for new therapies even more difficult. If available, patients with MCS should be considered potential candidates for clinical trials. It is crucial to continue research on the pathogenesis of MCS to improve diagnosis and treatment and find new potential drug targets for future therapies. The TKI and DR5 agonist, along with PI3K, MEK, HDAC, and CDK inhibitors, should be considered for future trials.

## Figures and Tables

**Figure 1 cancers-15-04581-f001:**
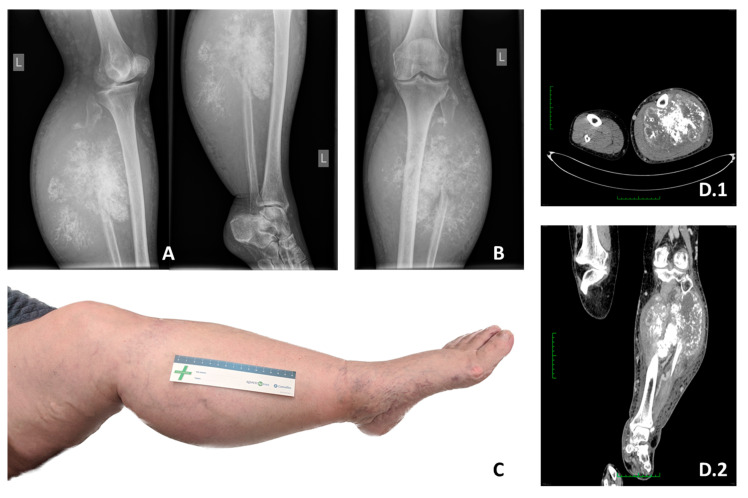
X-ray images showing locally advanced high-grade mesenchymal chondrosarcoma with abundant calcification and destruction of the proximal fibula. (**A**) Tibia Fibula Lateral X-ray view. (**B**) Tibia Fibula AP X-ray view. (**C**) Locally advanced high-grade mesenchymal chondrosarcoma of the left fibula in a 56-year-old woman for which the patient underwent transfemoral amputation. Axial (**D.1**) and Coronal (**D.2**) CT views depicting the size of the tumor.

**Figure 2 cancers-15-04581-f002:**
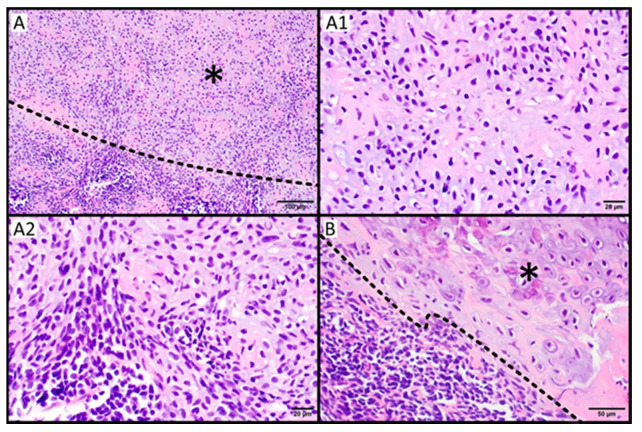
Mesenchymal chondrosarcoma. (**A**) Mesenchymal chondrosarcoma comprises round, undifferentiated cells and fields of cartilage tissue (asterisk) (**A1**). Islands of well-differentiated hyaline cartilage (**A2**). The round cell component could be the only finding in small biopsy material, which is challenging in diagnostics (**B**). Chondrocytes and the matrix may mimic osteoid disposition (asterisk). Scale bar: 100 μm (**A**); 20 μm (**A1**,**A2**); 50 μm (**B**).

**Figure 3 cancers-15-04581-f003:**
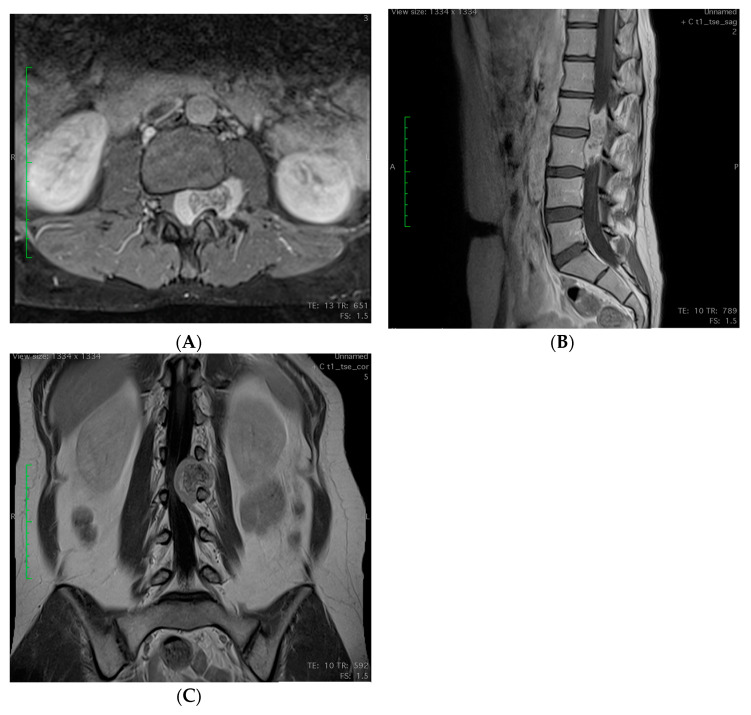
Lumbar spinal MRI showing an intraspinal extradural L2–L3 mesenchymal chondrosarcoma in a 31-year-old female. (**A**) Axial T1 Turbo Spin Echo (TSE) with fat suppression. (**B**) Sagittal T1 TSE. (**C**) Coronal T1 TSE.

**Table 1 cancers-15-04581-t001:** Potential therapies for selected genetic alterations and related pathways identified in mesenchymal chondrosarcoma.

Genetic Alteration Identified in Mesenchymal Chondrosarcoma	Related Processes and Activated Pathways	Potential Therapy	Clinical Status of Chondrosarcoma	Examples of Approved Application	References
HEY1-NCOA2 fusion	PDGF/PI3K/AKT	PI3K/AKT inhibitors	TQB3525 for advanced bone sarcomas with PI3KA mutations (NCT04690725)	Alpelisib in treatment of HR+/HER2− metastatic breast/Capivasertib in treatment of prostate cancer and solid tumors	[49,50]
	chromatin remodeling	HDACi	Belinostat in combination with guadecitabine or cedazuridine for locally advanced, metastatic and unresectable chondrosarcoma (NCT04340843)	Belinostat in in treatment of soft tissue sarcoma; Romidepsin in treatment of sarcoma et al.	[51]
	antiapoptotic activity	BCL2 inhibitors	-	Venetoclax used in treatment of hematological malignancies	[52]
*EWSR1-FLI1* fusion	chromatin remodeling	FACT inhibitor drug, e.g., CBL0137	-	-	[53]
loss of *CDKN2A*/p16	pRB pathway	CDK inhibitors	Abemaciclib for bone and soft tissue sarcoma, including chondrosarcoma, with CDK pathway alteration (NCT04040205)	Palbociclib 1, Ribo-ciclib 2 and Abemaciclib 3 in treatment of ER+/HER2− breast cancer	[54]
*PTCH1* point mutations	Hh pathway	Hh inhibitors	Vismodegib for advanced chondrosarcoma (NCT01267955)	Vismodegib in treatment of basal cell carcinoma treatment	[55]
*INSR* point mutations	Ras/MAPK	MEK inhibitors	-	Cobimetinib Selumetinib, or Binimetinib, Trametinib in treatment of melanoma or non-small-cell lung cancer	[56]

Abbreviations: hairy/split-enhancer-related with a fusion of the YRPW motif 1 (HEY1), nuclear receptor co-activator 2 (NCOA2), Ewing Sarcoma Breakpoint Region 1 (EWSR1), friend leukemia integration 1 transcrip-tion factor (FLI1), cyclin-dependent kinase inhibitor 2A (CDKN2A), patched protein homolog (PTCH1), insulin receptor (INSR1), platelet-derived growth factor receptor (PDGFR), phosphatidylinositol 3-kinase (PI3K), phospho-protein kinase B (AKT), retinoblastoma protein (pRB), Hedgehog (Hh), mitogen-activated protein kinase (MAPK), histone deacetylase inhibitors (HDACi), B-cell lymphoma (BCL2), facilitates chromatin transcription (FACT) complex, cyclin-dependent kinase (CDK), mitogen-activated protein kinase kinase (MEK).

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
