# Peer review of "Mesenchymal Chondrosarcoma from Diagnosis to Clinical Trials"

_cancers, 2023, doi:10.3390/cancers15184581_

Round 1

Reviewer 1 Report

I read with interest the Review Mesenchymal Chondrosarcoma from diagnosis to clinical trials . 

The Introduction is clear,  with important  epidemiological information.

It could be interesting to add some data on incidence and prevalence in different part of the world of this rare tumor, if available.

The chapter Clinical presentation  is well written. No changes

Imagining : the text is good. The  Figure 2 is not exemplificative: either the Xray or the  MRI  can  be attributed to MCS as well as to a Classical bone  CS. It should be changed with a most tipical figure of MCS.

 The Histopathology chapter is well written ,  no changes requested.

On the other Genetic alteration   is too long and without any practical   consequence, at least at present.

In my opinion a table resuming  all the  findings related to the   genetic alterations and  the  potential   therapies can be more useful and less prolix..

The chapter on Radical surgical treatment  is concise and clear . The phrase "  Intralesional resection provides good neurological outcomes but poor local control" is difficult to understand. In sarcoma surgery since the great lessons  of Prof Enneking ( 1981)  an intralesional resection is always  considered inadequate and  dangerous mainly  in spinal  localization.

 Chapter 5 is not local recurrence treatment, but local recurrence prevention.and can be inserted in the previous chapter  ( surgical treatment)  or  in the following ( radiation therapy).

No data are reported on the application of protontherapy.

Chapter 7 is about  treatment of locally advanced disease, but it     is  a mix of different clinical situations: neoadjuvant, adjuvant, locally advanced  and the final impression is a great confusion : MCS is not  so sensitive   to chemotherapy. and the specific role and position of Radiotherapy and chemotherapy is not clear ( concomitant? sequential?  chemotherapy before or after RT ?) 

The followig chapter  "systemic therapy in metastatic setting" is well  constructed and developed  The literature revision is complete and the bibliography is wide.  

Targeted therapies and immunotherapy is  just a suggestion: none of the drug gave a demonstrated activity .  The followig chapter ( clinical trials ) has the same sound  of inefficacy and can be included. into the previous one.

The conclusions are unsatisfactory. As matter of fact they resume  what has been previously  described in the  article.

In general the conclusions express the opinion  of the Authors  about the topics and  propose a future fields of research.

Author Response

Reviewer 1

I read with interest the Review Mesenchymal Chondrosarcoma from diagnosis to clinical trials . The Introduction is clear,  with important  epidemiological information.

Thank you for your kind remark.

It could be interesting to add some data on incidence and prevalence in different part of the world of this rare tumor, if available.

The incidence in Afroamericans and Asian population has been listed in introduction now.

The chapter Clinical presentation  is well written. No changes

Thank you for your kind remark.

Imagining : the text is good. The  Figure 2 is not exemplificative: either the Xray or the  MRI  can  be attributed to MCS as well as to a Classical bone  CS. It should be changed with a most tipical figure of MCS.

The dana on MRI and CT in mesenchymal chondrosarcoma was up-dated with data on typical presentation in these imaging methods to justify the figure.

 The Histopathology chapter is well written ,  no changes requested.

Thank you for your kind remark.

On the other Genetic alteration  is too long and without any practical consequence, at least at present.

Thank you for this remark. According to the fact that not much is known about pathogenesis of chondrosarcoma and mesenchymal subtype we wanted to summarize the current state of knowledge in this field and propose potential targets, we have added examples of known practical applications for proposed therapies and related drugs, and  some clinical trials’ data for chondrosarcoma.

In my opinion a table resuming  all the  findings related to the   genetic alterations and  the  potential   therapies can be more useful and less prolix..

Thank you for the suggestion. The table titled “Table 1. Potential therapies for selected genetic alterations and related pathways identified in mesenchymal chondrosarcoma.” has been added at the beginning of this chapter.

The chapter on Radical surgical treatment  is concise and clear . The phrase " Intralesional resection provides good neurological outcomes but poor local control" is difficult to understand. In sarcoma surgery since the great lessons  of Prof Enneking ( 1981)  an intralesional resection is always  considered inadequate and  dangerous mainly  in spinal  localization.’

Thank you for this remark, the sentence has now been corrected to make clear that such procedure is not appropriate. Additional data is cited to support the statement.

Chapter 5 is not local recurrence treatment, but local recurrence prevention.and can be inserted in the previous chapter  ( surgical treatment)  or  in the following ( radiation therapy).

Thank you for this comment. We moved this part to surgical treatment chapter.

No data are reported on the application of protontherapy.

Data on protontherapy has been added to radiotherapy section.

Chapter 7 is about  treatment of locally advanced disease, but it     is  a mix of different clinical situations: neoadjuvant, adjuvant, locally advanced  and the final impression is a great confusion : MCS is not  so sensitive   to chemotherapy. and the specific role and position of Radiotherapy and chemotherapy is not clear ( concomitant? sequential?  chemotherapy before or after RT ?) 

Treatment schedule has been clarified and appropriate guidelines were cited.

The followig chapter  "systemic therapy in metastatic setting" is well  constructed and developed  The literature revision is complete and the bibliography is wide.  

Thank you for your kind remark.

Targeted therapies and immunotherapy is  just a suggestion: none of the drug gave a demonstrated activity .  The followig chapter ( clinical trials ) has the same sound  of inefficacy and can be included. into the previous one.

Both chapters have been combined as drugs in development

The conclusions are unsatisfactory. As matter of fact they resume  what has been previously  described in the  article. In general the conclusions express the opinion  of the Authors  about the topics and  propose a future fields of research.

Thank you for this comment we have corrected conclusions and added some summary as below:

“In summary, there are no established guidelines for treatment of patients with advanced and metastatic MCS. Data of other therapeutic option, including  immuno-therapy efficacy MCS are limited and not successful enough. The rarity of these tumors makes the search for new therapies even more difficult. If available, patients with MCS should be considered potential candidates for clinical trials. It is crucial to continue re-search on  pathogenesis of MCS to improve diagnosis and treatment and find new potential drug targets for future therapies. The TKI and DR5 agonist, along with PI3K, MEK, HDAC, and CDK inhibitors should be considered for future trials.”

Reviewer 2 Report

In their manuscript entitled „Mesenchymal Chondrosarcoma from diagnosis to trials”, Dudzisz-Sledz and colleagues compile the current literature on the rare sarcoma entity mesenchymal chondrosarcoma (MCS). The authors present a thorough overview on prevalence, diagnostics, genetic alterations and therapeutic options like radical surgery, radiotherapy, chemotherapy, immunotherapy and additional ongoing trials.

The manuscript is relevant for clinicians closely related to the field of sarcoma diagnosis and therapy. I have only minor comments on the manuscript and recommend publication after revisions by the authors.

1) l. 44: the abbreviation CCS is used for the first time here, but not explained - this follows in l. 46-77. Please adjust.

2) Regarding subchapter 2.1.: as far as I see, information in l. 74 - 76 and l. 78 - 80 are somehow redundant. The same seems to be the problem with l. 70 -72 and l. 82 - 86. Similar problems can be found throughout the manuscript. I suggest proofreading the manuscript to sharpen some passages, there applicable.

3) l. 110 - 111: this statement is too unspecific. I would suggest to add some clarification.

4) Regarding subchapter 3.1.: I suggest including and discussing the work of Tanaka et al., JCI Insight 2023 (HEY1-NCOA2 expression modulates chondrogenic differentiation and induces mesenchymal chondrosarcoma in mice) in this subchapter, as it is - to my opinion - relevant to the chapter.

5) l. 346 -  p = 0.000 is a questionable representation of a highly significant association. Although I know that this is cited from the reference, I would recommend to change it to p < 0.001.

6) While most parts of the manuscript are interesting and well-written, the quality of subchapter 7.2 and 7.3 is limited. First, as it is two to three times stated, there seem to be no clinical trials including MCS, therefore the relevance of the two subchapters to the topic of the review is limited. Second, some information are redundant (for instance l. 536 - 538). And at last, it is not clear, how 7.3. should benefit for the topic. Although some information are useful (l. 545 - 549), both chapters are too long, and should be fused and sharpened.

7) Spelling and grammar is mostly fine. Some typos I found: l. 98: mesenchymal sarcoma à MCS; l. 312/313 - with localized disease; should be fixed in the final proofreading.

Finally, I want to thank the authors for sharing this interesting review with the scientific community. Best regards.

English is mostly fine - minor spellchecking required.

Author Response

1) l. 44: the abbreviation CCS is used for the first time here, but not explained - this follows in l. 46-77. Please adjust.

Thank you for this remark. It has been changed and explained in the first used of CCS abbreviation.

2) Regarding subchapter 2.1.: as far as I see, information in l. 74 - 76 and l. 78 - 80 are somehow redundant. The same seems to be the problem with l. 70 -72 and l. 82 - 86. Similar problems can be found throughout the manuscript. I suggest proofreading the manuscript to sharpen some passages, there applicable.

Chapter 2.1 has been re-written.

3) l. 110 - 111: this statement is too unspecific. I would suggest to add some clarification.

 Unspecific data has been removed

4) Regarding subchapter 3.1.: I suggest including and discussing the work of Tanaka et al., JCI Insight 2023 (HEY1-NCOA2 expression modulates chondrogenic differentiation and induces mesenchymal chondrosarcoma in mice) in this subchapter, as it is - to my opinion - relevant to the chapter.

Thank you for this suggestion. It has been added in this section as follow:

“In the latest study by Tanaka et al. treatment with the HDACi – Panobinostat – caused suppression of MSC cells growth both in vitro and in vivo. Panobinostat downregulate the expression of genes downstream of HEY1-NCOA2 and Runx2 [48].”

5) l. 346 -  p = 0.000 is a questionable representation of a highly significant association. Although I know that this is cited from the reference, I would recommend to change it to p < 0.001.

Thank you for this remark. It has been changed according to your suggestion.

6) While most parts of the manuscript are interesting and well-written, the quality of subchapter 7.2 and 7.3 is limited. First, as it is two to three times stated, there seem to be no clinical trials including MCS, therefore the relevance of the two subchapters to the topic of the review is limited. Second, some information are redundant (for instance l. 536 - 538). And at last, it is not clear, how 7.3. should benefit for the topic. Although some information are useful (l. 545 - 549), both chapters are too long, and should be fused and sharpened.

Data on completed trials has been cited now. Both in advanced disease section as well in new section on drugs in development. Chapters 7.2. and 7.3 are combined and supplemented with new data.

7) Spelling and grammar is mostly fine. Some typos I found: l. 98: mesenchymal sarcoma à MCS; l. 312/313 - with localized disease; should be fixed in the final proofreading.

Thank you for this remark. It has been corrected.

Finally, I want to thank the authors for sharing this interesting review with the scientific community. Best regards.

Round 2

Reviewer 1 Report

After the requested  revisions the paper is appropriate  and can be published